# Reducing Enteric Methanogenesis through Alternate Hydrogen Sinks in the Rumen

**Prasanta Kumar Choudhury** [1,2], **Rajashree Jena** [1,2], **Sudhir Kumar Tomar** [1] **and Anil Kumar Puniya** [1,*]

1   Dairy Microbiology Division, ICAR-National Dairy Research Institute, Karnal 132001, India
2   Department of Dairy Technology, School of Agricultural and Bioengineering, Centurion University of Technology and Management, Paralakhemundi 761211, India
*   Correspondence: akpuniya@gmail.com

**Abstract:** Climate change and the urgent need to reduce greenhouse gas (GHG) emission from agriculture has resulted in significant pressure on the livestock industry for advanced practices that are environmentally more sustainable. Livestock is responsible for more than 15% of anthropogenic methane ($CH_4$) emission via enteric fermentation and improved strategies for mitigating enteric $CH_4$ production therefore represents a promising target to reduce the overall GHG contribution from agriculture. Ruminal $CH_4$ is produced by methanogenic archaea, combining $CO_2$ and hydrogen ($H_2$). Removal of $H_2$ is essential, as its accumulation inhibits many biological functions that are essential for maintaining a healthy rumen ecosystem. Although several other pathways occur in the rumen, including reductive acetogenesis, propionogenesis, nitrate, and sulfate reduction, methanogenesis seems to be the dominant pathway for $H_2$ removal. Global warming is not the only problem associated with the release of $CH_4$ from ruminants, but the released GHG also represent valuable metabolic energy that is lost to the animal and that needs to be replenished via its food. Therefore, reduction of enteric $CH_4$ emissions will benefit not only the environment but also be an important step toward the efficient production of high-quality animal-based protein. In recent decades, several approaches, relying on a diverse set of biological and chemical compounds, have been tested for their ability to inhibit rumen methanogenesis reliably and without negative effects for the ruminant animal. Although many of these strategies initially appeared to be promising, they turned out to be less sustainable on the industrial scale and when implemented over an extended period. The development of a long-term solution most likely has been hindered by our still incomplete understanding of microbial processes that are responsible for maintaining and dictating rumen function. Since manipulation of the overall structure of the rumen microbiome is still a significant challenge targeting key intermediates of rumen methanogenesis, such as $H_2$, and population that are responsible for maintaining the $H_2$ equilibrium in the rumen could be a more immediate approach. Addition of microorganisms capable of non-methanogenic $H_2$ sequestration or of reducing equivalents are potential avenues to divert molecular $H_2$ from methanogenesis and therefore for abate enteric $CH_4$. However, in order to achieve the best outcome, a detailed understanding of rumen microbiology is needed. Here we discuss some of the problems and benefits associated with alternate pathways, such as reductive acetogenesis, propionogenesis, and sulfate and nitrate reduction, which would allow us to bypass $H_2$ production and accumulation in the rumen.

**Keywords:** hydrogenotrophy; methanogenesis; propionogenesis; reductive acetogenesis; sulfate reduction



## 1. Introduction

The rumen harbors a highly diverse and complex mixture of microorganisms, including archaea ($10^8-10^9$/mL), bacteria ($10^{10}-10^{11}$/mL), ciliate protozoa ($10^6$/mL), and fungi ($10^6$/mL), which facilitate the degradation of complex plant carbohydrates into small molecules [1] and ultimately provide metabolites that can be used by the ruminant animal [2–5]. Livestock are mainly fed with agricultural crops, which via microbial activity

are converted to metabolic intermediates (i.e., volatile fatty acids (VFAs), such as acetate, butyrate and propionate, and hydrogen ($H_2$) and gaseous end products such as carbon dioxide ($CO_2$) and methane ($CH_4$) [6]. Increased microbial $H_2$ production and its subsequent accumulation, which can be promoted by a high-starch diet, have several detrimental effects on the rumen ecosystem and that can be attributed to a decrease in rumen pH triggered by starch fermentation. These effects include the deactivation of specific biomass-degrading enzymes from some of the most efficient fiber degraders of the rumen microbiome but also system-level responses, such as the reduction of feed conversion within the rumen [7,8]. Methanogens, a group of microbes belonging to the phylogenetic group of the archaea, combine molecular $H_2$ with $CO_2$ to produce $CH_4$ during methanogenesis, enabling the removal of $H_2$ from the system [9,10]. Although this removal of $H_2$ is important for maintaining a healthy rumen ecosystem, from the viewpoint of nutrient expenditure methanogenesis is a costly process, accounting for a gross energy intake loss of 2–12% in ruminants [11–14]. Since the annual production of enteric $CH_4$ accounts for ~15% of total anthropogenic $CH_4$ emissions [11,15], with $CH_4$ having a global warming potential 23-fold higher than that of $CO_2$, there is also a real and severe environmental cost associated with the energy of the enteric $CH_4$ that is released into the atmosphere.

Strategies and factors for $CH_4$ abatement have been reviewed in the past [1,9,12,16–25] and many of the strategies used to mitigate $CH_4$ from ruminants involve the use of antibiotics, ionophores [26], halogenated $CH_4$ analogues [27–29], heavy metals [30], lipid-rich materials such as coconut oil [31–33], probiotics [27], bacteriocin [34], and numerous chemicals [35,36]. Immunization against methanogens [37,38], elimination of ciliate protozoa (defaunation) both in in vivo and in vitro [39] and addition of acetogenic bacteria to rumen fluid [40–42] in in vitro experiments have also been tested. Use of toxic chemicals and antibiotics as inhibitors, although considered an option in the past, are no longer accepted due to rising concerns regarding their impact on the environment, the animal, and potentially on the consumer of the animal products [43]. Interventions using phage therapy, altering methanogenic diversity and chemogenomic approaches [6] are some of the more recent technologies, but the extent to which these processes remove and eliminate the produced $H_2$ still remains to be investigated. Therefore, a critical step for a successful $CH_4$ reduction strategy may be one that uses natural processes within the rumen. One such approach relies on establishing a non-methanogenic sink for $H_2$ produced during fermentation. This review will focus on these $H_2$ elimination pathways.

## 2. Hydrogen: A Key Player in Rumen Fermentation

$H_2$ concentration plays a major role in the regulation of microbial fermentation in the rumen [44–47]. The partial $H_2$ pressure is a key regulator of $H_2$ metabolism and the fate of ruminal $H_2$ disposal with dissolved $H_2$ gas and $H_2$ ion determining the redox potential of the rumen liquor. The efficient elimination of $H_2$ enhances fermentation by reducing its inhibitory effect on microbial growth and microbial degradation of plant material [48,49]. Destiny of $H_2$ liberation is associated with favorable thermodynamic changes and an inverse correlation between Gibbs free energy ($\Delta G^0$) and the minimum partial $H_2$-pressure that is required for a reaction to continue: a reaction is considered to be thermodynamically more competitive when its requirement of $H_2$ partial pressure is low [50]. Due to this central regulatory role in rumen fermentation, $H_2$ can be considered to be the currency of ruminal fermentation [51]. Removal of the major fraction of the rumen $H_2$ occurs via the methanogenic archaea to $CH_4$, during which four moles of $H_2$ are consumed and converted into one mole of $CH_4$, which is then released into the atmosphere though eructation. During hydrogenotrophic methanogenesis, methanogens use $CO_2$ as carbon source and terminal electron acceptor and $H_2$ as electron donor. Other non-methanogenic rumen microbes, using $CO_2$ and other electron acceptors such as sulfate, nitrate, and fumarate, compete with methanogens for $H_2$, but they play a less dominant role in the removal of $H_2$ from the rumen ecosystem [52,53]. Non-methanogenic bacteria that use $H_2$ as electron donor include acetogens that reduce $CO_2$ to form acetate by the Wood-Ljungdahl pathway [54],

sulfate-reducing bacteria (SRB) that reduce sulfate to hydrogen sulfide [55], nitrate-reducing bacteria (NRB) that reduce nitrate ($NO_3$) to ammonia ($NH_4$) and fumarate-reducing bacteria that use $H_2$ to form succinate [56,57]. Succinate can subsequentially be decarboxylated to propionate, which is a valuable nutrient for the ruminant animal [58], either by the succinate producer itself or it can be transferred to succinate users as an intercellular electron carrier [45]. Figure 1 summarizes the microbial pathways for $H_2$ removal from the rumen.

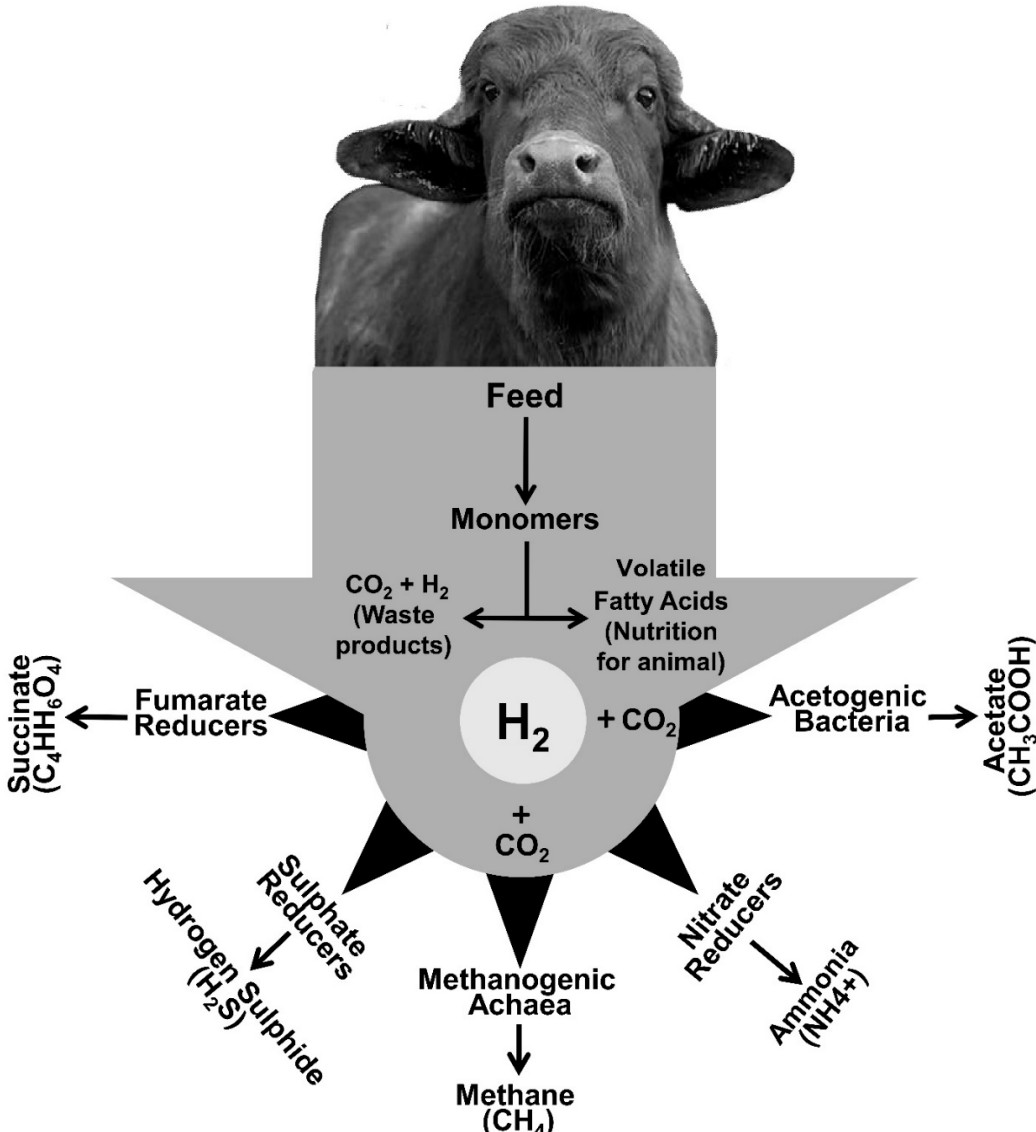

**Figure 1.** Major and minor $H_2$ and $CO_2$ sequestering pathways in rumen.

## 3. Alternative $H_2$ Sinks

The strategies to reduce $CH_4$ emission from enteric fermentation by non-methanogenic sinks are reviewed with respective mechanisms of action, thermodynamic changes; microorganism's involved, associated problems and anticipated management strategies are discussed (Table 1).

### 3.1. Reductive Acetogenesis

During reductive acetogenesis, also known as the Wood-Ljungdahl pathway or reductive acetyl-CoA pathway [7] $H_2$ and $CO_2$ are sequestered into acetate yielding energy for the ruminant host [19,54,59–63]. Due to the favorable energetics as well as absence of produced byproducts, reductive acetogenesis is a desirable way to eliminate excess $H_2$, and

$H_2$ concentration plays a vital role in deciding the fate of $H_2$ disposal. Acetogenesis can be autotrophic or heterotrophic, depending upon the type of substrate that is used. During autotrophic acetogenesis, two moles of $CO_2$ are reduced by four moles of $H_2$ to produce one mole of acetate ($4H_2 + 2CO_2 \rightarrow CH_3COOH + 2H_2O$) [60], whereas in heterotrophic acetogenesis, also referred to as homo-acetogenesis, one mole of hexose is converted to three moles of acetate, which is formed in a ratio of 2:1 from the oxidation of pyruvate and reduction of $CO_2$, respectively [64]. It is assumed that both autotrophic and heterotrophic acetogenesis occur simultaneously in the ruminant ecosystem. The Wood-Ljungdahl pathway has been described in diverse microbial ecosystems [59,65,66] where $H_2$ acts as an electron donor and $CO_2$ as an electron acceptor. The pathway contains two branches, methyl (western) and carbonyl (eastern) for synthesis of acetyl-CoA. The methyl branch is folate dependent where $CO_2$ reduced through formate and finally to methyltetrahydrofolate, while in carbonyl branch $CO_2$ reduced by carbon monoxide dehydrogenase to acetyl Co-A [62] (Figure 2). The change in Gibbs free energy during reductive acetogenesis is nearly $-10.2$ kJ/mol, while for methanogenesis from the same substrates is $-68.3$ kJ/mol [67]. This explains why reductive acetogenesis plays a minor role as hydrogenotrophic sink in the rumen when compared to microbial methanogenesis [68,69].

**Table 1.** $H_2$ sequestration sinks, involved mechanisms, associated problems and future directions.

| Categories | Sub Groups | End Products | Microbes (Examples) | Overall Reaction | $\Delta G^0$ (kJ) | Problems Associated | Management Strategies | Reference(s) |
|---|---|---|---|---|---|---|---|---|
| Methanogenic sinks | Methnogenesis | Methane ($CH_4$) | *Methanobrevibacter ruminantium, Methanomicrobium mobile, Methanobacterium bryantii, Methanobrevibacter smithii, Methanosarcina barkeri, Methanoculleus olentangyi* | $4H_2 + CO_2 \rightarrow CH_4 + 2H_2O$ | $-134.0$ | Source of ruminal $CH_4$, but not desirable as potent GHG. | Releases $H_2$ accumulation in rumen and need to be suppressed. | [2,4–6,11,12,18, 20,22–25,45,63] |
| Non-Methanogenic Sinks | Sulfate Reduction | Hydrogen Sulfide ($H_2S$) | *Desulfovibrio desulfuricans, D. vulgaris, Desulfatomaculum* spp. | $4H_2 + 2H^+ + SO_4 \rightarrow H_2S + 4 H_2O$ | $-234.0$ | Undesirable reaction in rumen owing to toxicity of $H_2S$. | Most energy efficient sink in rumen dietary level and feeding strategy must taken into account. | [53,55,70–75] |
| | Reductive acetogens | Acetic acid ($CH_3$-COOH) | *Eubacterium limosum, Acetitomaculum ruminis, Blautia* spp, *Clostridium* spp., *Peptostreptococcus productus, Ruminococcus schinkii, Clostridium difficile* | $4H_2 + 2CO_2 \rightarrow CH_3COO^- + H^+ + 2H_2O$ | $-71.6$ | Desirable, but needs high levels of $H_2$ partial pressure. | Alteration of rumen microflora with a low $H_2$ threshold possessing capacity for reductive acetogenesis. | [7,10,19,47,54, 66,76,77] |
| | Nitrate Reduction | Ammonia ($NH_4$) | *Selenomonas ruminantium, Veillonella parvula* and *Wolinella succinogenes* | $4H_2 + 2H^+ + NO_3^- \rightarrow NH_4^+ + 3H_2O$ | $-519.0$ | Undesirable reaction in rumen owing to possible accumulation of toxic nitrite. | Gradual adaption of animal to supplement used and development of favorable microflora. | [53,67,70,78–91] |
| | Propionogenesis | Propionic acid ($CH_3CH_2COOH$) | *Fibrobacter succinogenes, Selenomonas ruminantium* ssp. *ruminantium, Selenomonas ruminantium* ssp. *lactilytica, Veillonella parvula* and *Wolinella succinogenes* | $C_6H_{12}O_6 + 2H_2 \rightarrow 2CH_3CH_2COOH + 2H_2O$ | $-84.0$ (Fumarate to succinate) | Desirable reaction, but required substrate is costly. | Balancing minimum level in diet and dosing desired microbes governing propionate synthesis. | [57,92–103] |

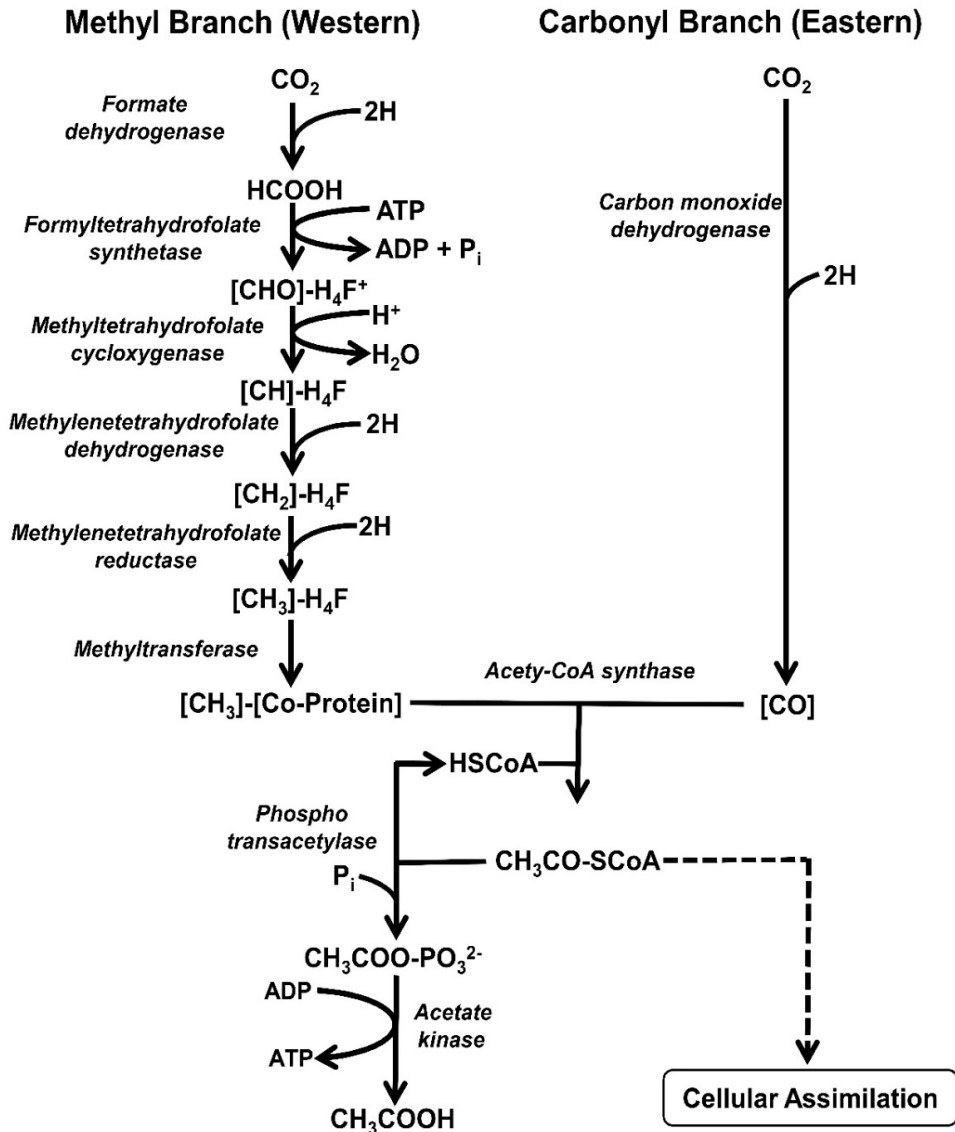

**Figure 2.** The Wood-Ljungdahl pathway of reductive acetogenesis.

Ruminal acetogens are not obligate in their substrate specificity and can contribute to $H_2$ production rather than $H_2$ consumption [41,104,105]. López et al. [106] reported that acetogenic bacteria can consume $H_2$ and $CO_2$ to form acetate significantly in the rumen when methanogenesis is inhibited. In the same study, they reported that an increase in the number of acetogenic bacteria cannot compete with methanogens. LeVan et al. [40] observed enhancement of in vitro reductive acetogenesis in incubations when methanogenesis was inhibited by BES with addition of rumen acetogen *Acetitomaculum ruminis* 190A4 and concluded that both selective inhibition of methanogenesis and addition of acetogens are crucial for the prevailing reductive acetogenesis under $H_2$ limiting conditions. However, Joblin [59] reported that ruminal acetogens dominate over methanogens and reduce $CH_4$ emission in vitro even if at low concentrations $H_2$. Competition for $H_2$ exists in rumen where acetogens are dominant hydrogenotrophs in the early rumen microbiota and methanogens replace them in later stage [61,76]. Fonty et al. [77] reported that reductive acetogens can maintain a functional rumen and replace methanogens as a sink for $H_2$ in methanogen free lambs and contributed to 21 to 25% to the rumen fermentation in vivo. However, Gagen et al. [76] observed in lambs that methanogen colonization does not significantly alter acetogen diversity isolated after 17 h after birth. Inhibition of ruminal methanogenesis and dosing of acetogens may lead significant increase in reductive ace-

togenesis [40,41]. Mitsumori et al. [107] reported a change in acetogen diversity in vivo in Holstein steers fed an antimethanogenic compound bromochloromethane (BCM). In another study, lambs removed from their mothers within 2 days of birth and kept in isolation appeared to use more metabolic $H_2$ via reductive acetogenesis and less $CH_4$ than conventionally raised lambs [108]. Other strategies may involve acetogen "enhancers" to provide acetogens with an advantage over methanogens, for example through the addition of yeast cells. *Saccharomyces cerevisiae* was reported to stimulate ruminal acetogens and their use of $H_2$ even in the presence of a methanogen in vitro [109]. Similarly, Yang et al. [110] observed enhanced acetogenesis and $H_2$ use increased the efficacy of acetogens in the presence of *S. cerevisiae* TWA4 strain.

In other gut environments, reductive acetogenesis is the major $H_2$ removal pathway and may be a useful source of potential acetogens to compete successfully with methanogens in the rumen [66,69,104,111,112] and thermodynamic control is not the single aspect for regulation of methanogen-acetogen interactions [69,104]. The prevailing $H_2$ gradient and the ability to grow mixotrophically in these environments may give acetogens a competitive advantage [67,113]. Acetogens isolated from eastern gray (*Macropus giganteus*), red kangaroos (*Macropus rufus*) and tammar wallaby (*Notamacropus eugenii*) have the capacity to compete with methanogens [111,112]. Methanogenesis was inhibited to an undetectable limit in reactors simulating the human gut by the addition of *Peptostreptococcus productus* [42]; interestingly this effect was diminished with ruminal fluid incubations [41]. A comparative analysis of acetogen isolated from ruminants (Ser5, Ser8, CA6 and SA11), marsupials (YE255, YE257 and YE266) and two reference isolates (*Acetitomaculum ruminis* and *Eubacterium limosum*) for $H_2$ use and acetate production showed that marsupial isolates (YE255, YE257 and YE266) are more efficient in using $H_2$; than ruminal isolates (CA6 and SA11) followed by reference isolates of acetogens [114]. Efficacy of acetogens to compete ruminal methanogenesis is observed to be source and strain dependent. Therefore, microbes with competitive ability at low $H_2$ partial pressure and/or addition from low $CH_4$ emitting animals must be taken into consideration for a fruitful reductive acetogenesis to establish in livestock.

### 3.2. Sulfur Reduction

Sulfate reduction is a thermodynamically highly favored process for $H_2$ removal in the rumen system [67]. Sulfate reducing bacteria (SRB) can be categorized based on the process they are employing for sulfate reduction into either assimilatory or dissimilatory SRB [115,116] (Figure 3). Both groups exist in rumen and facilitate the reduction of sulfur to hydrogen sulfite $HSO_3^-$ and hydrogen sulfide $H_2S$. Dissimilatory reduction of sulfur compounds is used for energy generation, whereas during assimilatory sulfur reduction, sulfur compounds are incorporated into biological molecules that are necessary for cell survival [116,117]. In the formation of hydrogen sulfide, four moles of $H_2$ are consumed for each mole of $H_2S$ generated. Energetically 1 ATP is consumed in the dissimilatory reduction of sulfate process to produce sulfide, whereas during the assimilatory process two ATP are used without generating $H_2S$. Dissimilatory reduction is the key route of sulfate metabolism in the rumen [118].

Dissimilatory SRB are strict anaerobic mesophiles, mostly Gram-negative, rod-shaped bacteria [119] that are ubiquitous in the digestive tract of mammals [55,120,121]. Members of the Desulfovibrionaceae (i.e., members of the genus *Desulfovibrio*) are the dominant SRB in ruminants [122,123]. Other abundant SRBs have been identified as belonging to the genus *Desulfotomaculum* and *Fusobacterium* [119,124]. Inclusion of SRB and/or sulfate in ruminant diet has shown to reduce $CH_4$ emission and enhance digestibility of the feed (Table 2). The recommended concentration of sulfur in growing beef cattle diet is 0.15% [125] and 0.14 to 0.26% in growing lamb diet [126]. Ruminant diets deficient in sulfur are connected with decreased microbial protein synthesis, digestibility, and lactate use [127,128]. Whanger and Matrone [129] reported microorganisms from sulfur-deficient animal contents could not synthesize butyrate and higher VFAs from acetate. Improved dry

matter (DM) digestion, rumen fermentation and bacterial population in sheep fed a high sulfur diet were reported [130]. Patterson and Kung [131] observed adding sulfur at 0.3% DM improved cellulose digestion threefold in in vitro fermentations. Limited availability of sulfate in rumen has a direct effect on $H_2$ pressure in the rumen by suppressing microbial $H_2$ consumption and dietary increasing their concentration supports lowering $CH_4$ production. Change in microbial biomass and particularly an increase in SRB was observed with sulfate supplementation of the ruminant diet [70] and dissimilatory sulfate reduction was found to be proportional and also limited to the amount of sulfur available. Supplementation of sulfur to the regular diet fed to goats and lams resulted in the reduction of enteric $CH_4$ [70,71]. Paul et al. [124] reported supplementation of sulfate reducing bacteria SRBBR5, a strain capable of sulfate reduction, which resulted in the decrease $CH_4$ emissions from 2.66 to 1.64 mM $CH_4$/g DM truly digested after 72 h of fermentation without affecting methanogens and fungal population. In the same study, digestibility was reported to be increased significantly (15% in apparent digestibility and 40% in true digestibility), whereas $H_2S$ concentration remained unaffected. Similarly, Wu et al. [72] reported that increased sulfur content of the animal diet resulted in decreased $CH_4$ emission (12.54 vs. 5.11 μM), total gas production ((39.1 vs. 27.1/mL culture), digestibility (63.0% vs. 51.5%)) and concentration of total VFAs in vitro, while increasing ammonia with no significant effect on archaea population.

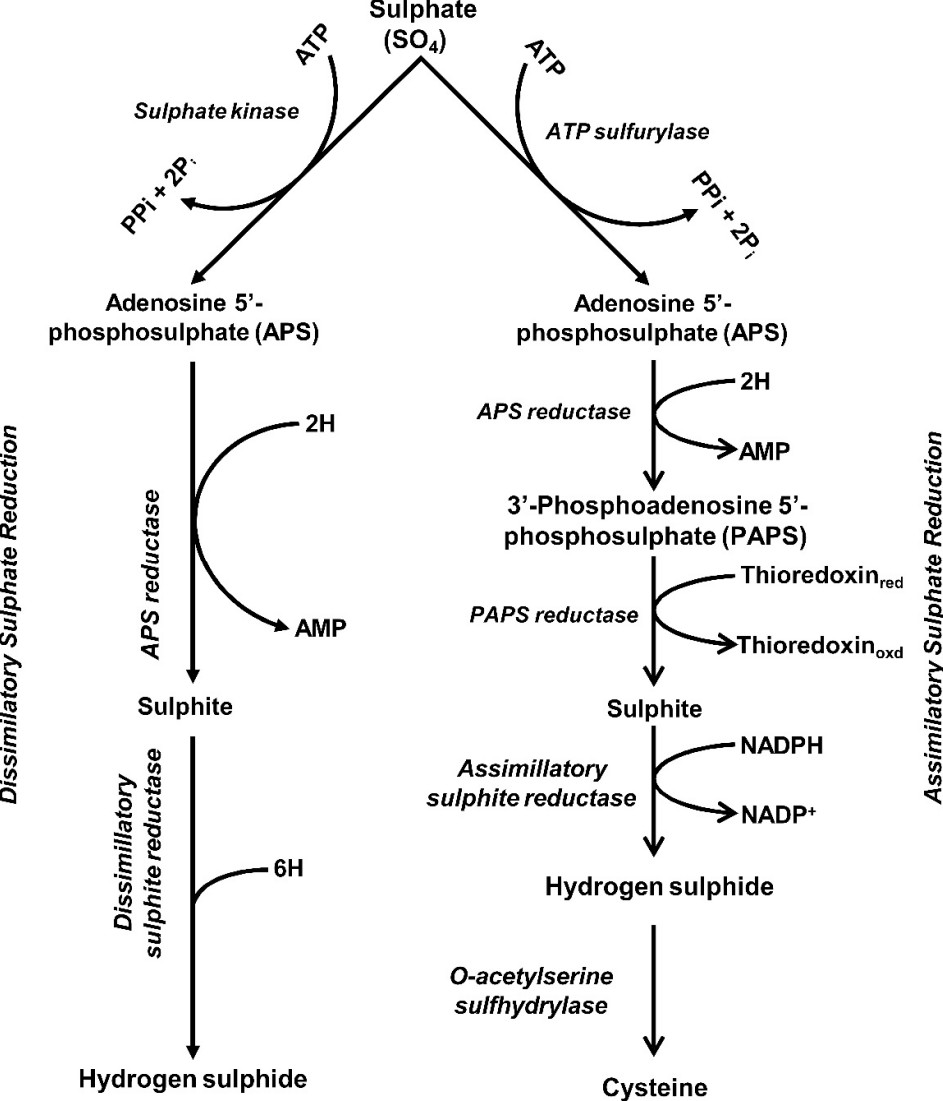

**Figure 3.** Dissimilatory and assimilatory route of sulfur reduction in rumen.

**Table 2.** Dietary sulfur, nitrate, fumarate and/or combinations on CH$_4$ production in in vitro or animal trials.

| Dietary Supplements | Source and Level | Model | CH$_4$ Reduction (%) | References |
|---|---|---|---|---|
| Sulfur | Sulfate (2.6%) | Sheep | 16 | [70] |
| | Sodium sulfate (0.8%) | Goat | 14.2 | [71] |
| Nitrate | Pottasium nitrate (4%) | Sheep | 23 | [90] |
| | Pottasium nitrate (5%) | Cattle | 43 | [132] |
| | Pottasium nitrate (6%) | Cattle | 27 | [133] |
| | Nitrate (22 g/kg DM) | Cattle | 32 | [88] |
| | Nitrate (2.6%) | Sheep | 32 | [70] |
| | Calcium ammonium nitrate (2.84%) | Cattle | 41 | [134] |
| | Sodium nitrate (1.3 g/kg BW) | Sheep | 50.4 | [87] |
| | Nitrate (21 g/kg DM) | Cattle | 16 | [89] |
| | Calcium nitrate (3.8%/DM) | Goat | 23.2 | [71] |
| Fumarate | Fumaric acid (2% DM) | Cattle | 23 | [96] |
| | Encapsulated fumarate (10%) | Sheep | 76 | [135] |
| | Sodium fumarate (400 µM) | In vitro | 17 | [94] |
| | Sodium fumarate (500 µM) | In vitro | 60 | [136] |
| | Fumarate (3.5 g/L) | In vitro | 38 | [137] |
| | Sodium fumarate (6.2 mM) | In vitro | 17 | [95] |
| | Fumaric acid (8% DM) | Sheep | 12 | [138] |
| | Sodium acrylate | In vitro | 8 | [94] |
| | Sodium fumarate | In vitro | 17 | [94] |
| | Fumarate (10 mM) | In vitro | 17 | [139] |
| | Fumarate (30 mM) | In vitro | 11 | [140] |
| Combinations | Sulfur (2.6%) + Nitrate (2.6%) | Sheep | 47 | [70] |
| | Sodium sulfate (0.8%) + Calcium nitrate (3.8%) | Goat | 34.9 | [71] |
| | Sodium nitrate (1.3 g/kg BW) + GOS | Sheep | 52.9 | [87] |
| | Sodium nitrate (1.3 g/kg BW) + Nisin (3 mg/kg BW) | Sheep | 56.3 | [87] |
| | Sodium nitrate (5%) + Sulfur (0.4%) | Sheep | 19.6 | [141] |
| | Sodium nitrate (5%) + Sulfur (0.4%) | Goat | 18.2 | [141] |

However, application of sulfur to the ruminant diet has some serious problems, especially when performed under not closely monitored conditions, since sulfur concentrations above a critical dose result in the H$_2$S, which is toxic to the host animal [53,72–74,142]. H$_2$S has limited solubility and is readily absorbed through the rumen wall into the blood stream [143], and therefore interrupts animal performance. The appearance of polioencephalomalacia (PEM) or cerebro-cortical necrosis occurs when sulfide travels through the blood to the brain, leading to death and contributes substantial economic loss to livestock industry [142]. Other associated problems of increased H$_2$S concentrations include adverse effect on the activity of respiratory enzymes (e.g., catalases, peroxidases, carbonic anhydrase, dopa-oxidases, dehydrogenases, and dipeptidases, cytochrome-c oxidase), production of sulfhemoglobin, depressed rumen motility, decreased mineral use, and several adverse effects on oxidative metabolism and energy generation in animals [115,144]. There have been reports of approaches that address and solve these H$_2$S toxicity problems. For example, feedlot cattle responded well to diets high in sulfate ferric citrate decrease [145] and passive immunization targeting SRB [146] adverse H$_2$S toxicity. However, significantly more work in this area is needed and efficient strategies to overcome H$_2$S toxicity still need to be identified before sulfur/sulfate can be considered to be a viable feed additive to inhibit methanogenesis.

### 3.3. Nitrate Reduction

The mechanism of nitrate (NO$_3$) reduction (Figure 4) can serve as an alternate pathway for lowering CH$_4$ emission due to NO$_3$ with a higher affinity for H$_2$ than CO$_2$ [78,147]. In anaerobic systems, nitrate reduction occurs by three distinct mechanisms [147], dissim-

ilatory nitrate reduction to nitrogen gas (denitrification), assimilatory nitrate reduction (respiratory nitrate reduction, ANR) to ammonia and dissimilatory nitrate reduction to ammonia (Figure 4). Denitrification proceeds in a stepwise manner, in which nitrate ($NO_3$) is reduced to nitrite ($NO_2$), which then is reduced to nitric oxide (NO), which is further reduced to nitrous oxide ($NO_2$), and eventually to nitrogen gas ($N_2$). Although denitrification does not play a major role in the rumen under normal physiological conditions, trace amounts of nitrogen oxide can be measured when nitrate is abundant. In assimilatory nitrate reduction, the product of the enzymatic reaction remains in the organism itself to enable microbial protein synthesis. Nitrate is reduced to nitrite by NADH reduction reactions and nitrite is reduced to ammonia by respiratory ammonification, coupled with ATP production [79] to fulfill the energy needs associated with this form of nitrate reduction. In contrast to the energy producing dissimilatory nitrate reduction, high ammonia concentrations have an inhibitory effect on ANR. Hence ANR plays no major function on the rumen environment, where ammonia is abundant and rumen microorganisms will use this pathway to primarily synthesize sufficient ammonia to meet their requirements for biosynthesis and storage [148]. In this category the microorganism *Wolinella succinogenes* is the most comprehensively studied organism that carries out respiratory nitrite ammonification [79]. The genome sequences of *Wolinella succinogenes* showed that it is a close relative of the pathogenic epsilon proteobacteria *Helicobacter pylori* and *Campylobacter jejuni* and the first non-pathogenic bacteria whose genome sequence was determined [149].

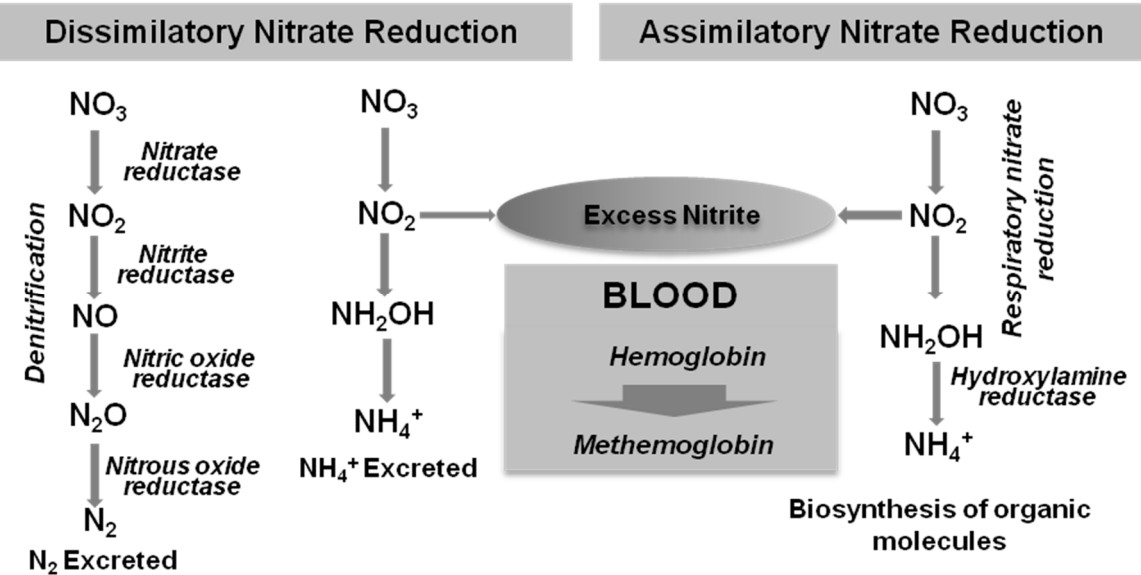

**Figure 4.** Assimilatory and dissimilatory routes of nitrate reduction.

Dissimilatory nitrate reduction to ammonia is the predominant pathway of nitrate metabolism in the rumen [80,81]. The conversion of nitrate to ammonia is thermodynamically more favorable to methanogenesis [67,81,150]. The reduction of nitrate to nitrite following reduction of nitrite to ammonia yields more Gibbs free energy than the reduction of $CO_2$ to $CH_4$ [67]. These processes could be the major route of $H_2$ elimination if sufficient nitrate is available in an actively fermenting rumen ecosystem. The conversion of nitrate to ammonia consumes eight reducing electrons and each mole of nitrate that is reduced could theoretically lower $CH_4$ production by one mole [80]. High organic matter concentration, low redox potential, and presence of sulfide in the rumen favors dissimilatory reduction [151], which is also not inhibited by high concentration of ammonia. The generated ammonia will be available for microbial biomass synthesis and provides an important supply of fermentable nitrogen [152]. The possibility of nitrate as an alternative $H_2$ sink to $CO_2$ in ruminant is somehow problematic and requires a detailed understanding of the rumen microbiome and the microbial processes involved in nitrate and nitrite metabolism

due to the formation of toxic intermediates [147]. In the rumen, the nitrate conversion rate is higher than the rate of the subsequent conversion of nitrite to ammonia [139]. Excess nitrate consummation by the ruminant leads to the accumulation of nitrite which can trigger methemoglobinemia [82] and have adverse effects on the oxygen transport system due to oxidation of ferrous ($Fe^{2+}$) to the ferric ($Fe^{3+}$) yielding a stable oxidized form of hemoglobin (methemoglobin; MetHb) which is unable to release oxygen to the tissues. In mild cases, increases levels of MetHb can lower animal performance, but in severe cases this can be lethal [153].

A controlled and supervised administration of nitrate and nitrate-reducing bacteria into the rumen has been used as a successful strategy to allow the rumen and its microbiome to acclimatize to increasing levels of nitrate and enhance their ability to reduce nitrite [154,155]. Introducing microorganisms that have nitrite reductase activity and therefore an advantage over methanogenic archaea when competing for $H_2$ [156] ultimately affects ruminal $CH_4$ production and nitrite reduction [78,157]. *Veillonella parvula*, *Selenomonas ruminantium* and *Wolinella succinogenes* reduce nitrate and nitrite for example have been shown to be promising probiotcis that can be added to the rumen ecosystem to alleviate high nitrate concentrations in ruminant feed [83,158]. In another study, the occurrence of nitrate in diet controls nitrate-reducing bacteria *Wolinella succinogenes* and *Veillonella parvula* in medium containing ground hay and concentrate were estimated by competitive PCR [84]. Simon [79] studied the administration of *Wolinella succinogenes* having the ability to convert nitrate to ammonia with minimum nitrite accumulation in in vitro studies. A similar effect was also established in vivo using *Escherichia coli* strains with high nitrate/nitrate reductase activity [157]. The addition of anaerobic cultures of *E. coli* W3110 or *E. coli* nir-Ptac with nitrate in cultures of mixed ruminal microorganisms decrease nitrite toxicity and $CH_4$ production in vitro [159,160]. Sakthivel et al. [78] found that decreased $CH_4$ formation and enhanced nitrate and nitrite removal from ruminal digesta in the presence of a nitrate-reducing rumen bacterium (unidentified) in vitro. Nitrate supplementation linearly increased total VFA concentration and cellulolytic bacteria species (*Ruminococcus flavefaciens*, *Ruminococcus albus* and *Fibrobacter succinogenes*) in rumen-fistulated steers [85]. In the same study, they reported that *Campylobacter fetus*, *Selenomonas ruminantium*, and *Mannheimia succiniciproducens* were major nitrate-reducing bacteria in steers and their number linearly increased with level of nitrate supplementation [85]. Prebiotics also affect nitrate reduction and supplementation of galacto-oligosaccharide (GOS) decreased nitrite accumulation and up to 11% reduction in $CH_4$ emission [134,161].

Rumen protozoa have been reported to accelerate nitrate reduction when co-cultured with bacteria [162]. A significant proportion of nitrate reduction in the rumen with higher acetate to propionate ratio was observed with protozoa fraction in vitro [163]. van Zijderveld et al. [70] reported that though the number of protozoa remained unaffected with dietary supplementation of nitrate, yet a decline in protozoal count (60%) was observed with nitrate administration in the rumen [157]. Asanuma et al. [86] reported sevenfold declined protozoal population in goats adapted to dietary nitrate and significant decrease in the population of methanogen, protozoa, and fungi with increase in *S. bovis* and *S. ruminantium* was observed with nitrate supplementation. As methanogenic archaea are associated with protozoa surfaces, decrease in their number may have a crucial role in reducing $CH_4$ emission. Overall, inadequate information regarding the shift of rumen microbiome, in response to diet limits the challenges to reduce nitrate/nitrite reduction in rumen. Ruminant diet and substrate type also influence the reduction rate of nitrate and nitrite. In rumen bacterium *S. ruminantium* increase nitrate reductase per cell mass was reported in the presence of nitrate [164]. Higher reduction rate was observed on lactate as compared to glucose, and further enhanced with succinate. Regularly dosing nitrate directly into the rumen lowered $CH_4$ production was observed in vivo [87,165].

Leng [80] reported that a host of factors *viz.* fermentable carbohydrate, adequate sulfur level, low soluble protein fraction and a source of bypass protein favor the use of nitrate and lower $CH_4$ production. Hulshof et al. [88] found 32% decrease in $CH_4$ production in

steers when fed nitrate at 2.2% of DM. van Zijderveld et al. [70] reported feeding nitrate or sulfate had no effect on the concentration of short chain fatty acids in rumen fluid after 24 h of feeding. However, the molar proportion of branched-chain VFAs varied, higher when sulfate and lower when nitrate fed diet were administered in lambs at a proportion of 2.6% of dry matter each in corn silage-based diet for 28 days. A significant decrease in $CH_4$ production was at maximum immediately after nitrate feeding, but the effect was uniform for the entire day in sulfate feeding. An increase in nitrate level in diet was accompanied by a linear decrease in $CH_4$ reduction [89]. Leng [80] observed inclusion of 1% potassium nitrate in a diet decreased $CH_4$ production by 10%. Decrease in $CH_4$ production by 23% in sheep was achieved with oat hay diet supplemented with 4% potassium nitrate compared to control diet made iso-nitrogenous by the addition of urea [90]. Nitrate supplementation has also been proposed to be a useful non-protein nitrogen (NPN) source for ruminants and as a replacement for urea [88,89,132,133,141,166]. Numerous in vivo and in vitro studies confirmed the efficacy of feeding nitrate on decreasing enteric $CH_4$ emissions without resulting in clinical signs of toxicity [70,89,90,150,166]. In an experiment when rumen fluid from a Jersey bull was incubated with sodium nitrate (12 mM) in vitro, 70% reduction of $CH_4$ level with 30% decrease in gas production was achieved [167]. Zhou et al. [91] further reported complete inhibition of $CH_4$ production with nitrate level more than 12 mM. The use of nitrate appears to be one of the improved strategies to be adapted in livestock sector to reduce enteric $CH_4$ fermentation, but the animals need to be acclimatized to nitrate feeding by step by step increasing the level in the diet to avoid harmful effects.

*3.4. Propionogenesis*

Redirection of metabolic $H_2$ away from $CH_4$ toward volatile fatty acids, primarily propionate, has been suggested to be an efficient strategy to reduce enteric $CH_4$ production in vivo, while also increasing the animal's feed efficiency [92,102,103,138,168,169]. The limiting factor for propionogenesis (Figure 5) as $H_2$ sink and a mean of consistently lowering the partial $H_2$ pressure in the rumen is substrate availability [63,170]. Formation of propionate can occur through either the microbial fumarate-succinate pathway [171] or the microbial conversion of pyruvate to lactate and acrylyl-CoA ester and the subsequent reduction in propionate [172]. Ellis et al. [50] reported that reduction of fumarate to succinate is thermodynamically more favorable than methanogenesis within the physiological partial $H_2$ pressure of the rumen that needs required substrate are availability in rumen.

In accordance with the notion that propionogenesis is a substrate-limited process, inclusion of propionate precursors or dicarboxylic acids in the diet shifted rumen fermentation toward propionic acid production and decreased $CH_4$ yield [93,94,173] (Table 2). The addition of fumaric acid for example yielded reduced in vitro and in vivo $CH_4$ production [53,91,94–98,140,174–177] and Wallace et al. [178] also observed an increase in weight gain in lambs fed when fumaric acid was added to the fed. Increased DM digestibility with decreased $CH_4$ production with addition of sodium fumarate in vitro was also observed [95]. Itabashi et al. [179] marked fumaric acid fed together with salinomycin to Holstein steers increased molar concentration of propionic acid resulting in a 16% decrease in $CH_4$ production, suggesting that the addition of ionophores together with fumarate might have a synergistic effect of these compounds on $CH_4$ production. In another study, steers fed with fumaric acid (2% DM) on a sorghum silage-based diet was reported to reduce $CH_4$ by 23% [96]. Asanuma et al. [140] suggested fumarate to be an economical feed additive for reduced $CH_4$ production. García-Martínez et al. [175] reported that effects of fumarate on rumen fermentation depend on the nature of the incubated substrate and significant response was observed with high forage diet. Wood et al. [135] reported $CH_4$ emission in sheep reduced to approximately 76% with fumaric acid (10%) encapsulated in fat. Demeyer and Henderickx [136] found 60% inhibition of $CH_4$ production by addition of fumarate (500 μM) in vitro. Similarly, a 17% decrease in $CH_4$ production in response to the addition of 400 μM fumarate was observed [94]. In another study, 38% reduction in $CH_4$ production in continuous fermenters was recorded with fumarate supplementation [137].

Ungerfeld et al. [176] recommended that low concentration of fumaric acid would be more effective in reducing CH₄ production. Beauchemin and McGinn [180] observed fumaric acid caused potential valuable changes in ruminal fermentation but no measurable reductions in CH₄ emissions. McGinn et al. [26] also reported that fumaric acid had no effect on CH₄ emissions, in growing beef cattle. Although the majority of the evidence support fumaric acid addition in animal diet, yet economic arguments and acidosis problems restrict their application in animal feed.

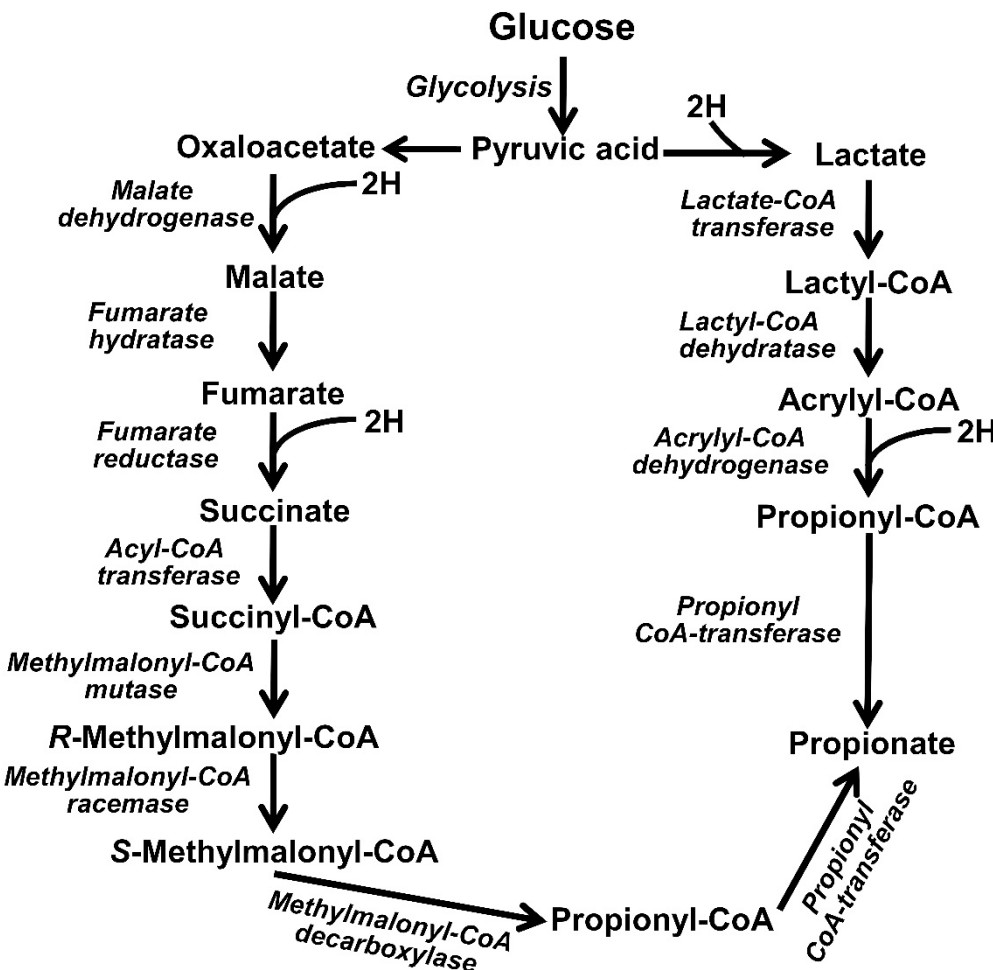

**Figure 5.** Pathway for propionate synthesis from oxaloacetate and lactate.

Similarly, the addition of malate, a key intermediate of the inverse citric acid cycle and of the succinate propionate pathway, has also been intensively studied for its ability to stimulate propionate production in the rumen [173,181,182]. Feed supplemented with malate (140 g/day) in lactating dairy cows reported an increase in milk production and feed conversion efficiency [183]. Dosing malate in ruminant diet increased nitrogen retention in sheep and steers while it also improved average daily weight gain and feed efficiency in calves [184]. In experiments, changes in rumen pH, VFA profiles and decreased CH₄ production analogous was also noticed when malate was added to diet [181,185,186]. Carro et al. [186] reported that although malate decreased CH₄ production per unit of DM digestion, but enhanced fiber digestibility resulted net increase in CH₄ production. Foley et al. [99,100] noticed little benefit gained from the dietary supplementation of malate in dairy cows. However, Carro and Ranilla [187,188] observed that malate beneficially affected in vitro rumen fermentation, with decreased CH₄ production and L-lactate concentrations.

In addition to the direct inclusion of propionate intermediates, several studies investigated the use of probiotics to enhance propionate production [57,93,101,140]. *Veillonella parvula*, *S. ruminantium* subsp. *ruminantium*, *S. ruminantium* subsp. *lactilytica* and *Fi-*

*brobacter succinogenes* have been reported to support propionate production and a high capacity to $CH_4$ reduction [189]. In another study, in vitro addition of fumarate-reducing bacteria *Mitsuokella jalaludinii* increased succinate production with significant decrease in $CH_4$ production and change in rumen microbial diversity was reported [190]. Nisbet and Martin [174] observed 10 mM-fumarate stimulated the growth of *S. ruminantium* in pure cultures. López et al. [95] found a significant increase in cellulolytic bacteria with addition of 7.35 mM-fumarate to semi-continuous fermenters. Similarly, Zhou et al. [101] observed addition of disodium fumarate inhibited the growth of methanogens, protozoa and fungi while cellulolytic bacteria (*R. albus*, *F. succinogenes* and *B. fibrisolvens*) and proteolytic bacteria (*B. fibrisolvens*, *P. ruminicola*, and *Clostridium* sp.) showed positive response. Compositional changes in the bacterial population in goat's rumen and improved metabolism of rumen lactate fermentation were also reported with addition of disodium fumarate [97]. In majority addition of fumarate and malate as well as microbe capable of reducing malate or fumarate seem to be effective in controlling ruminal methanogenesis and escalating supported microflora but dosing level and cost needs to be considered for this approach.

Addition of lactic acid bacteria (LAB) and their metabolites to enhance propionate synthesis for the $H_2$ sequestration also been invested by many researchers [141,191–193]. LAB stimulated the growth of lactate using microbes resulted in increased propionic acid production and leading to a substantial decrease in the $H_2$ availability for $CH_4$ production [141]. *Lactobacillus pentosus* D31 was reported to reduce $CH_4$ production (13%) over a period of 4 weeks dosed with $6 \times 10^{10}$ cfu to each animal every day [194]. Bacteriocins, the metabolites of LAB reported to decrease $CH_4$ production with promising results both in vitro and in vivo experiments. The possibility of employing bacteriocins for $CH_4$ mitigation from streptococci of rumen origin has recently been reviewed [66]. Nisin a bacteriocin produced from *Lactococcus lactis* was observed to decrease 36% $CH_4$ production in vitro [195]. In sheep a 10% decrease in $CH_4$ emissions (g/kg DMI) with nisin supplementation was reported [196]. Similarly, bovicin HC5 [34] and pediocin [197] was shown to decrease $CH_4$ production by 53% and 49% in in vitro trials, respectively. LAB used as silage inoculants [198] also reported $CH_4$ diminishing (8.6%) activities and possible increase in propionic acid (4.8%). Reduced gas production by silage treated groups compared with the untreated silage evoked a shift in fermentation [199]. Similarly, in another experiment carried out by Cao et al. [200] with vegetable residue silage, there was a decrease in $CH_4$ production (46.6% reduction) and increase in in vitro dry matter digestibility. Huyen et al. [201] reported that LAB strains are most promising when used as silage inoculants and observed to decrease $CH_4$ production and increase DM digestibility. In addition to that increase in cellulolytic microorganisms and decrease in CH production using corn stover silage inoculated with *Lactobacillus plantarum* and increase in lactic acid fermentation, in vitro digestibility and $CH_4$ mitigation in the forage sorghum mixture silages was also observed using *Lactobacillus casei* TH14 inoculant [202]. These experiments showed that LAB, their metabolites, and applications in silage have a positive effect in decreasing $CH_4$ yield possibly by stimulating the lactate-propionate pathway or release of inhibitory compounds.

## 4. Conclusions and Future Prospects

Ruminal methanogenesis, contributing $CH_4$ to the atmosphere, is directly and inversely linked to the animal productivity. The ability to control $CH_4$ emission especially reduce methanogenesis from agriculture has enormous environmental and socioeconomic implication, but it also requires a detailed understanding of the microorganisms and microbial processes that are involved. Although a complete understanding of these highly interwoven microbial and metabolic networks has still not been achieved and most likely will not be feasible in the immediate future, there are some aspects that are reasonably well understood. These aspects represent a promising starting point for targeted $CH_4$ reduction from ruminants. One of the promising key intermediates that has been recognized as such

and that has received significant attention for targeted $CH_4$ mitigation is metabolic $H_2$ and the metabolic pathways, microbes and enzymes involved its production and consumption.

Since $H_2$ is an immediate precursor for the archaeal reduction of $CO_2$ into $CH_4$, biological approaches that redirect $H_2$ away from archaeal methanogenesis and into alternate metabolic pathways seem to be the most promising approaches to convert feed carbon into metabolic energy for the ruminant instead of releasing it into the atmosphere. Redirecting $H_2$ through reductive acetogenesis and propionogenesis has advantages over other pathways due to production of valuable metabolic end products that can be used by the host animal as nutrients and can be converted into animal proteins for human consumption. Although our understanding of how to redirect metabolic $H_2$ into more favorable pathways facilitates the production of value-added metabolic intermediates and therefore redirects otherwise lost feed energy, several issues related to the fine tuning of this redirection, such as the co-factor requirements, toxicity of metabolic intermediates, as well as thermodynamics of competing metabolic processes, need to be investigated in greater detail. A further aspect that will have to be investigated further and that will have direct implications for the translational value of findings on the area of rumen nutrition and function is the link and dependence of the rumen microbiome and its function in dietary conversion. With recent advances in omics technologies and the foray into the metabolic processes that are actually engaged in the rumen microbiome under certain physiological conditions, we now have the tools that will enable us to lay the foundation for a high-resolution picture of the rumen ecosystem and its microbial processes.

**Author Contributions:** P.K.C. conceived the idea for the article, prepared and edited the final manuscript. R.J. and P.K.C. prepared the figure, tables and edited the final manuscript. A.K.P. and S.K.T. help in final editing and participated in developing the idea and critically revised the manuscript. All authors have read and agreed to the published version of the manuscript.

**Funding:** This research received no external funding.

**Institutional Review Board Statement:** Not applicable.

**Informed Consent Statement:** Not applicable.

**Data Availability Statement:** Not applicable.

**Acknowledgments:** The authors would like to acknowledge for necessary facilities provided by ICAR-National Dairy Research Institute, Karnal, Haryana, India for literature collection of this review. Institutional fellowship received to PKC during the period of study is highly obliged.

**Conflicts of Interest:** The authors declare no conflict of interest.

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
