# Peer review of "Reducing Enteric Methanogenesis through Alternate Hydrogen Sinks in the Rumen"

_methane, doi:10.3390/methane1040024_

Round 1

Reviewer 1 Report

Dear Authors,

This review research article details the hydrogen sinks in the rumen by using the biochemical concepts with the evidences from the published works. This present work is well done and good at biochemistry for reducing methanogenesis. However, I feel that some points could be improved. Therefore, I comment this paper needs a revision prior to publication.

Thank you.

Comments:

Abstract

-          None.

Introduction

-          L44: Please remove the spaces around “/”.

-          L48: Please remove “(“ which put before i.e.,.

-          L57: Please adjust “hydrogen“ to “H2”.

-          L69: revise “ciliate protozoans” to “ciliate protozoa”.

-          L77: revise “lumen” to “rumen”.

2. H2…

-          L80: revise “:” in subscript form to the normal form.

-          L81: hydrogen = H2

-          L88: Gibbs Free Energy = Gibbs free energy

-          L105: In Figure 1, revise “Fatty Acids“ to “Volatile Fatty Acids”, “For” to “for”, “Succinic acid” to “Succinate”, and “Fumarate Reduciers” to “Fumarate Reducers”.

-          L106: revise the atomic numbers in H2 and CO2 to the subscript form.

3. Alternative H2 Sinks  

-          L131: Remove spaces around /.  

-          L134: Revise “TABLE” to “Table”.

-          L134: In Table 1, revise “Methano-genic” to “Methanogenic”; and the formats such as lower case or upper case, use or not use spaces around + and mole numbers. In addition, please use “.” at the end of each sentence.

-          L135: In Figure 2, revise “Kinase” to “kinase”.

-          L151: “Saccharomyces cerevisiae” is presented, in L158 the same name should be reduced to “S. cerevisiae”. Please keep this logic for all microbial names in this manuscript.

-          L167-168 vs L169: “marsupials (YE255, YE257 and YE266)” is used, but “marsupial isolates (ser5 and ser8)” is concluded, please correct.

-          L172: Edit spaces around /.

-          L175: “H2” adjust the atom number to subscript form.

-          L185: In Figure 3, revise “Kinase” to “kinase”, “Adenosine 5' Phosphosulfate” to “Adenosine 5'-Phosphosulfate”, “3′ Phosphoadenosine 5′ phosphosulfate” to “3′-Phosphoadenosine 5′-phosphosulfate”, and “Hydrogen Sulphide” in right hand side to “Hydrogen sulphide”.

-          L189 and L190: Remove “spp.”.

-          L192: Edit “0.14 to 0.26%” to “0.14% to 0.26%”.

-          L195: Revise “fatty acids” to “VFAs”.

-          L196: Edit “dry matter” to “dry matter (DM)”

-          L197: Verb “were able to show” is not suitable for this sentence. Please edit.

-          L204: There is a blank of space, please adjust.

-          L205: Change “2.66 to 1.64” to “1.64 to 2.66”.

-          L208: Please specify “12.54 to 5.11” or “12.54 vs. 5.11”.

-          L208-209: Edit “µMol” to “µM”, “((“ to “(“, “).)” to “)”, and “VFA” to “VFAs”,

-          L212: TABLE = Table; Dietary sulphur, nitrate and/or combinations = Dietary sulphur, nitrate, fumarate, and combinations; trials = trials.; Kg = kg; Sodium Nitrate = Sodium nitrate; remove spaces around /; Encapsulated Fumaric acid = Encapsulated fumarate; μm = μM; Percent CH4 Reduction = CH4 reduction (%).

-          L216: “there by” or “thereby”, please verify.

-          L226: “still need to me identified”?

-          L230: Edit “In anaerobic systems nitrate reduction…” to “In anaerobic systems, nitrate reduction…”.

-          L243-247: Sentence “The genome…” is too complex and long, please break down it into 3-4 small sentences.

-          L261: methaemoglobinaemia = methemoglobinemia (MetHb)?, please check.

-          L262: “hemoglobin (methemoglobin; MetHb)” = “methemoglobin” ?, please check.

-          L275: “by competitive PCR [74] was reported” = “by competitive PCR was reported [74]”

-          L278: remove spaces around /

-          L282: “TVFA” = “total VFAs”

-          L289: Revise “In rumen bacterium S. ruminantium increase in nitrate reductase…” to “S. ruminantium increase nitrate reductase…”.

-          L304: @ = at

-          L304 and L307: dry matter = DM

-          L306: VFAs ?

-          L311-313: In sentence “Decrease in CH4 production by 23% in sheep was achieved with oat hay diet supplemented with 4% potassium nitrate or made iso-nitrogenous by the addition of urea [130].”, do you mean either potassium nitrate or urea supplementation could decrease CH4 production by 23% in sheep? However, please revise for improving the readability.

-          L315-316: “[59, 97, 130, 172-173; 12]”, please correct for the order.

-          L323: hydrogen = H2

-          L331: In Figure 5, revise Reductase to reductase, and Mutase to mutase.

-          L337: DM?

-          L347 and L348:  µM?

-          L361: “what has been reported were observed”, please edit.

-          L363: digestion = digestibility

-          L367: additional or addition?

-          L367: Please add comma before several

-          L372: please correct “rumen diversity”, should it be rumen microbial diversity (?).

-          L374: 7·35 = 7.35

Conclusions and Future Prospects

-          L383: Please remove “annually” because enteric CH4 emission occurs every time.  

-          L384: Revise “The ability to control, and especially reduce, CH4 emission from…” to “The ability to control CH4 emission, especially reduce methanogenesis from…”

-          L404: composition = conversion

References  

-          Making italic for the special words, for example “in vitro” in L409.

-          Making lowercase form for research article topics, for example L557, L560, L597, L599,…

Additional comment

-          L59: Phase “accounting for an energy loss of 8-13% from the ingested fed” is not clear for it meaning. Please specify the energetic reference name and the type of data (range, or mean to maximum). The energy contents in feeds are gross energy (GE), digestible energy (DE), metabolizable energies, and net energies. Methane is a GE loss that occurs between DE and ME. If we use GE, the range 8-13% is not suitable for all ruminant diets. The minimum should be lower than this, for example about 2 or 3 %GE; then, the possible range is about 2 -14 %GE. In my opinion, a modern review article could descript methane eructation in both %GE and %DE, which you need more references.

-          In 3.4. Propionogenesis, L323-330 is aimed to introduce the scope of biochemistry in Figure 5, L333-355 has focused the effect and evidence of fumarate, L356-366 shows those of malate, and L367-381 displays those of propionic acid producing microbes (inoculants). Why authors ignore the effect and evidence of lactate? Please put the reason in the manuscript that why it should not be included, or alternatively insert a paragraph to evaluate lactate addition levels and rumen methanogenesis. Is there lactate utilizing microbes in the rumen? In many experiments, they evidenced that improving ensiling by adding lactic acid producing bacteria could increase lactic acid concentration in silage and modify rumen methanogenesis.

Author Response

Dear Reviewer

Methane Journal

Sub: Submission of the revised research article Methane-2030707

Dear Sir/Madam

Please find herewith the revised research articles on “Reducing Enteric Methanogenesis Through Alternate Hydrogen Sinks in the Rumen” of consideration for publication in “Methane”. The authors have gone through the entire manuscript and corrected all the queries raised by the reviewers in track change mode. For your reference an author’s response to reviewers’ comments has been attached.

Thanks

Reviewer 2 Report

This manuscript presents a nice review of various hydrogen consuming processes occurring within the rumen ecosystem with emphasis on their potent to compete against hydrogen-consuming methanogens. The manuscript is generally well written and well organized although there are some relatively minor English usage issues that disrupt the flow of some sentences.  I list below only a few minor comments for the authors to consider:

Line 56; suggest "to produce CO2..."

Line 61; suggest subscripting 2 on CO2.

Line 70; suggest writing "in in vitro incubations..." or "in vitro cultures..." or "in vitro experiments..." or something similar to improve flow.

Line 94; suggest writing "acceptor and use H2 as an electron donor..."

Table 1; there are some spacing issues within the table that may be fixed.

Line 137; suggest "specificity and can contribute to H2..."

Line 165; do you mean "inhibited to an undetectable limit...."?

Line 175; suggest subscript 2 in H2.

Line 204; there is an awkward space in this sentence.

Line 216 and surrounding; H2S can also be inhaled by the animal as ruminants do not burp as we do but their process of eructation has the animal eructating into the oral-nasal cavity whereby the gas is inhaled first and then then exhaled.

Line 238; The authors may want to double check their ANR description. It is my understanding that assimilative nitrate reduction is an energy consuming process done to reduce nitrate to ammonia which is then used to synthesis microbial protein.     

Author Response

(The authors gave the same response as above.)

Round 2

Reviewer 1 Report

Dear Authors,

Thank you very much for writing a good work. I feel that you need a minor revision. Please revise the manuscript as follows:

(1) all "in vitro" and "in vivo", including those in References, are italic

(2) in Table 2, use kg (not Kg), and

(3) in L381-397, more recent references (year 2021-2023) are required.

Thank you,

Author Response

(The authors gave the same response as above.)
